# In Vivo Administration of Recombinant Human Granulocyte Colony-Stimulating Factor Increases the Immune Effectiveness of Dendritic Cell-Based Cancer Vaccination

**DOI:** 10.3390/vaccines7030120

**Published:** 2019-09-19

**Authors:** Shigetaka Shimodaira, Ryu Yanagisawa, Terutsugu Koya, Koichi Hirabayashi, Yumiko Higuchi, Takuya Sakamoto, Misa Togi, Tomohisa Kato, Takashi Kobayashi, Tomonobu Koizumi, Shigeo Koido, Haruo Sugiyama

**Affiliations:** 1Department of Regenerative Medicine, Kanazawa Medical University, Uchinada, Kahoku 920-0293, Japan; koya@kanazawa-med.ac.jp (T.K.); taku0731@kanazawa-med.ac.jp (T.S.); m-togi@kanazawa-med.ac.jp (M.T.);; 2Center for Advanced Cell Therapy, Shinshu University Hospital, Matsumoto 390-8621, Japan; ryu@shinshu-u.ac.jp (R.Y.); kohira@shinshu-u.ac.jp (K.H.); 3Department of Clinical Laboratory Sciences, Shinshu University School of Medicine, Matsumoto 390-8621, Japan; sasa0922@shinshu-u.ac.jp; 4Shinshu Cancer Center, Shinshu University Hospital, Matsumoto 390-8621, Japan; takob@shinshu-u.ac.jp (T.K.); tomonobu@shinshu-u.ac.jp (T.K.); 5Department of Gastroenterology and Hepatology, The Jikei University School of Medicine, Kashiwa, Chiba 277-8567, Japan; shigeo_koido@jikei.ac.jp; 6Department of Cancer Immunology, Osaka University Graduate School of Medicine, Osaka 565-0871, Japan; sugiyama@sahs.med.osaka-u.ac.jp

**Keywords:** dendritic cell, cancer vaccine, vaccination, acquired immunity, granulocyte colony-stimulating factor, tetramer analysis

## Abstract

Significant recent advances in cancer immunotherapeutics include the vaccination of cancer patients with tumor antigen-associated peptide-pulsed dendritic cells (DCs). DC vaccines with homogeneous, mature, and functional activities are required to achieve effective acquired immunity; however, the yield of autologous monocyte-derived DCs varies in each patient. Priming with a low dose of recombinant human granulocyte colony-stimulating factor (rhG-CSF) 16–18 h prior to apheresis resulted in 50% more harvested monocytes, with a significant increase in the ratio of CD11c^+^CD80^+^ DCs/apheresed monocytes. The detection of antigen-specific cytotoxic T lymphocytes after Wilms’ tumor 1-pulsed DC vaccination was higher in patients treated with rhG-CSF than those who were not, based on immune monitoring using tetramer analysis. Our study is the first to report that DC vaccines for cancer immunotherapy primed with low-dose rhG-CSF are expected to achieve higher acquired immunogenicity.

## 1. Introduction

Despite significant advances in cancer therapy, such as surgical techniques, radiotherapy, and systemic therapy including immune checkpoint inhibitors [1,2,3,4,5,6], it remains extremely challenging to treat advanced cancers with organ involvement and distant metastasis.

Manufacturing technology for antigen-presenting cell (APC)-based immunotherapy is being developed using active dendritic cells (DCs), the most potent APCs of the immune system, for therapeutic vaccination against cancer. APC-based immunotherapy with active DCs has been reported for the induction of effective immunity against cancer antigens [7]. DCs expressing both PD-L1 and PD-1 can virtually interact with any PD-1 and PD-L1-positive cells, respectively, with suppressive activity on CD4^+^ and CD8^+^ T cells and promotive expression of CD4^+^CD25^+^FoxP3^+^ regulatory T cells [8]. Interference with the PD-1/PD-L pathway can increase the immunostimulatory properties of the DCs to activate T cells [8]. It is possible that the immune checkpoints on the DCs would interfere with the response of antitumor immunity. Immune DCs are generated from peripheral monocytes expressing tumor-specific antigens and have been applied in active immunotherapy against cancers [9,10], which requires large-scale ex vivo generation of homogeneous, mature, and functional DCs. Cancer vaccines containing autologous monocyte-derived mature DCs (mDCs) are conventionally manufactured using granulocyte-macrophage colony-stimulating factor (GM-CSF) and interleukin (IL)-4 and are principally targeted against a specific antigen. The US Food and Drug Administration approved sipuleucel-T (Provenge^®^, Dendreon Corporation, Seattle, WA, USA) as autologous DC-based immunotherapy for patients with metastatic hormone-refractory prostate cancer, providing a new personalized cancer immunotherapy treatment option [11]. DC vaccines targeting Wilms’ tumor 1 (WT1) have been previously shown to be safe and feasible with few adverse reactions in patients with advanced cancer, including colorectal cancer [12], pancreatic cancer [13,14], lung cancer [15], high-grade glioma [16], and pediatric cancer [17,18]. The efficacy of DC-based immunotherapy is more evidently demonstrated by the delayed separation of the survival curve with a benefit in terms of prolonged overall survival rather than conventional evaluation approaches such as the use of the response rates [13,15,19]. Therefore, DC-mediated cancer immunization with DCs of a validated quality could potentially yield a superior effect compared with the conventional approach.

Large-scale preparation of DC vaccines with homogeneous, mature, and functional profiles are a prerequisite for achieving efficacious cancer immunotherapy [20]. However, the factors directly or indirectly predictive of the DC yield in individual patients remain unelucidated. The yield of autologous DCs from monocytes obtained by apheresis varies from small to large. It was reported that a history of smoking affected the manufacturing quantity of individualized DCs for vaccination in patients with various cancer types [21]. Recombinant human granulocyte colony-stimulating factor (rhG-CSF) is widely used for the collection of progenitor cells from the peripheral blood for allogeneic transplantation and of granulocytes for transfusion and is based on its ability to mobilize these cells [22]. The administration of 75–100 µg of rhG-CSF for neutropenia caused by chemotherapy has been demonstrated to stimulate the proliferation of myeloid precursors and accelerate the release of neutrophils from the bone marrow [23,24,25].

The factors affecting the induction of tumor antigen-specific cytotoxic T cells (CTLs) using DC vaccination have yet to be fully elucidated. The objective of this study was to address the effectiveness of rhG-CSF administration and how it might affect the manufacture of DC-based vaccines for clinical use in cancer immunotherapy.

## 2. Materials and Methods 

### 2.1. Patients

Cancer patients were enrolled in the DC vaccination study after obtaining their informed consent. The application requirements and conditions for the DC vaccination study were described previously [12]. We administered add-on DC vaccination in combination with conventional chemotherapy for each patient. The DC vaccination study was approved by the Ethics Committee of Shinshu University School of Medicine [approval number 1199, December 2, 2008 (initial clinical study) and 2704, April 8, 2014 (consecutively followed approval of advanced medical care in Japan)], and all investigations were performed according to the Declaration of Helsinki. The patients enrolled for DC vaccination were retrospectively evaluated to identify those who had undergone rhG-SCF treatment 24–96 h prior to apheresis to treat neutropenia. A total of 108 apheresis sessions were conducted for 90 patients with advanced cancers, such as breast (17), colon/rectum (19), lung (12), ovary (11), pancreatic (33), and stomach (16) cancers, including repeated sessions in 18 patients; the number and phenotype of their DCs were evaluated. The number of monocytes was determined evaluable in 40 patients administered rhG-SCF (75 μg filgrastim; Kyowa Hakko Kirin Co., Ltd., Tokyo, Japan) 16–18 h prior to apheresis. CD34^+^ cells were simultaneously analyzed in 20 evaluable patients who received rhG-SCF.

There were 54 patients displaying human leukocyte antigen (HLA)-A*24:02, in whom immune monitoring using tetramer analysis was evaluated after one course of the DC vaccination study. These patients had cancer of the colon/rectum (12), lung (8), biliary tract (8), pancreas (18), and stomach (8). The first 15 patients were consecutively enrolled into the group without rhG-CSF priming, and then the next 43 patients were enrolled in the group that was treated with rhG-CSF 16–96 h before the apheresis session. Thirty-nine out of 43 patients who were primed with rhG-CSF (75 μg filgrastim) 16–18 h prior to apheresis included those with colon/rectum (10), lung (5), biliary tract (5), pancreatic (13), and stomach (6) cancers. In the untreated (control) group of 15 patients, the types of cancer included colon/rectum (2), lung (3), biliary tract (3), pancreatic (5), and stomach (2) cancers.

### 2.2. Manufacture of a DC Vaccine and Vaccination

mDCs were generated in compliance with Good Gene, Cellular, and Tissue-based Products Manufacturing Practice (GCTP) according to the Act on the Safety of Regenerative Medicine introduced in Japan on 25 November 2014 [26]. The final number of viable DCs was equivalently aliquoted as 1 × 10^7^ in each lot. Immature DCs were generated by culturing adherent cells in AIM V medium (Gibco Laboratories, Gaithersburg, MD, USA) containing 50 ng/mL GM-CSF (GENTAUR Belgium BVBA, Brussels, Belgium) and 50 ng/mL IL-4 (R&D Systems Inc., Minneapolis, MN, USA) for 5 days using mononuclear cell-rich fractions isolated via apheresis as previously described [12]. mDCs were differentiated from immature DCs by stimulation with OK-432 (10 μg/mL streptococcal preparation; Chugai Pharmaceutical Co., Ltd., Tokyo, Japan) and 50 ng/mL PGE2 (Daiichi Fine Chemical Co., Ltd., Toyama, Japan) for 24 h. The mDC products were cryopreserved at −152 °C or in the gas layer of a liquid nitrogen tank until the day of administration.

The antigenic profiles of the mDCs were determined as CD11c^+^, CD14^−^, HLA-DR^+^, HLA-ABC^+^, CD80^+^, CD83^+^, CD86^+^, CD40^+^, and CCR7^+^ using flow cytometry. The criteria for DC vaccine administration were as follows: purity (defined as >90% of CD11c^+^ CD14^−^ CD86^+^ HLA-DR^+^ cells), >80% viability, mDC phenotype, negative for bacterial and fungal infection after 14 days, presence of endotoxin ≤ 0.05 EU/mL, and negative for mycoplasma [12].

A modified WT1_235–243_ peptide (CYTWNQMNL) compatible with HLA-A*24:02 by substitution of the second amino acid (methionine, M) with tyrosine (Y) can induce CTLs to be more effective than the wild-type peptide [27,28,29]. For each vaccination, an aliquot of frozen mDCs was thawed immediately prior to clinical use and primed with 100 μg/mL of good manufacturing practice-grade WT1 peptide (NeoMPS Inc. San Diego, CA, USA) at 4 °C for 30 min, washed twice to remove any free peptides, and re-suspended in 1 mL of 1–2 Klinische Einheit (KE) of OK-432. The WT1 peptides were HLA-A*24:02-restriction modified (CYTWNQMNL, residues 235–243). One course of seven biweekly sessions was performed with 1–3 × 10^7^ DCs with 1–2 KE of OK-432 intradermally injected at bilateral axillar and inguinal areas per session in accordance with previously described protocols for clinical use as per GCTP [12,30].

### 2.3. Surface Marker Analysis of the Obtained mDCs

A BD FACSCalibur™ flow cytometer (BD Biosciences, San Jose, CA, USA) was used to detect the surface molecules expressed on the obtained DCs. The following antibodies were applied to viable DC populations, excluding lymphocytes, in the cryopreserved samples: fluorescein isothiocyanate (FITC)-labeled anti-human CD14 (clone 61D3; eBiocience, San Diego, CA, USA), CD40 (clone 5C3; eBiocience), CD80 (clone L307.4; BD Biosciences), HLA-ABC (clone W6/32; eBiocience), CD3 (clone SK7; BD Biosciences), phycoerythrin (PE)-labeled anti-human CD11c (clone B-ly6; BD Biosciences), CD83 (clone HB15e; eBiocience), CD86 (clone IT2.2; eBiocience), CD19 (clone 4G7; BD Biosciences), and HLA-DR (clone LN3; eBiocience). CD34^+^ cells were analyzed using FITC-labeled anti-human CD34 (clone 8G12; BD Biosciences) to assess the effect of priming with rhG-CSF (filgrastim) for 16–18 h.

### 2.4. Immune Monitoring with Tetramer Analysis

Next, to prove the hypothesis of an increase in acquired immunity using G-CSF-primed DC vaccines, we used tetramer analysis to assess the ratio and degree of the immune response to DC vaccination both with and without rhG-SCF 16–18 h prior to apheresis. Freshly isolated peripheral blood mononuclear cells were stained with PE-conjugated human immunodeficiency virus/HLA-A*24:02 tetramer as a negative control or with PE-conjugated WT1-modified peptide/HLA-A*24:02 tetramer (Medical & Biological Laboratories Co., Ltd., Nagoya, Japan). Other stains included allophycocyanin-conjugated anti-CD3 monoclonal antibody and FITC-conjugated anti-CD8 monoclonal antibody prior to analysis by flow cytometry on a BD FACSCanto^TM^ II (BD Biosciences). The presence of WT1 antigen-specific CTLs was defined according to the following criteria: (i) >0.02% WT1-positive CD8^+^ T cells in a population of 50,000–100,000 lymphocytes with no evidence of false-positive cells and (ii) WT1-positive population described as clustered and not diffuse [31].

### 2.5. Statistical Analysis

Spearman’s rank-order correlation was used to analyze the processes between apheresis of the monocytes and the manufacture of mDCs. Differences in the peripheral blood, apheresed monocyte counts, the number of DCs, and the DC ratio for each factor were determined via an unpaired *t*-test. A paired *t*-test was utilized to assess the normal distribution of quantitative differences between the numbers of monocytes and CD34^+^ cells in each patient treated with rhG-CSF 24 h prior to apheresis. The acquisition of WT1 antigen-specific CTLs in patients with HLA class-A*24:02 was analyzed in G-CSF-primed and non-primed groups to compare distributions and medians without following normal distribution using the Mann–Whitney *U*-test. The Chi-square test was used to qualitatively analyze the distribution of G-CSF-primed and non-primed patients according to the types of cancers. A *p*-value < 0.05 was considered statistically significant. All analyses were performed using IBM SPSS Advanced Statistics 23.0 (IBM Japan, Tokyo, Japan).

## 3. Results

### 3.1. Apheresis for DC Vaccination and DC Products

The total number of apheresed monocytes was dependent on the peripheral monocyte count on the day of the apheresis (Figure 1a; *r* = 0.828, *p* < 0.0001, *n* = 108). The statistical significance of the mDC product was dependent on the number of apheresed monocytes because the DC yield varied from a few DCs to a large amount, even when the standard operating procedure was applied at a single institute. Therefore, it was considered that some factors other than the number of apheresed monocytes also influenced the qualitative and quantitative yield of the DC product (Figure 1b; *r* = 0.435, *p* < 0.0001, *n* = 108). The mean number of mDCs and the viability of the cells prior to cryopreservation were 20.2 ± 9.6 × 10^7^ and 95.6% ± 4.3%, respectively (Figure 2a). The ratio of DCs to monocytes was 17.7% ± 7.7%.

### 3.2. Surface Marker Analysis of the Obtained DCs

The percentages of the yielded DCs with CD11c^+^CD14^−^ and HLA-ABC^+^DR^+^ phenotypes were 97.2% ± 6.7% and 97.4% ± 4.3%, respectively (Figure 2a). mDCs able to present tumor antigens to T cells were confirmed by the presence of surface markers CD80, CD83, and CD86 and HLA-ABC and HLA-DR antigens. The obtained DCs with the mature phenotype strongly expressed CD11c, CD40, and CD86 and HLA-ABC and HLA-DR antigens, with few CD14^+^ monocyte markers. The counts of CD80^+^ and CD83^+^ DCs varied among patients (Figure 2b).

### 3.3. rhG-CSF Administration Influencing Monocyte-derived DC Manufacture

rhG-CSF was administered subcutaneously 24–96 h prior to apheresis to patients who required it because of leukopenia that had developed during chemotherapy. The cut-off for rhG-CSF administration was defined either as neutrophil count lower than 500/μL or counts lower than 1000/μL with a risk of fever. There were no cases with adverse effects due to the subcutaneous administration of rhG-CSF during the chemotherapy regimens. The numbers of both peripheral and apheresed monocytes were higher within 24 h and within 24–96 h, respectively, in the G-CSF-treated groups than in the non-G-CSF group (Figure 3a). The DC yield was higher in the 24-h (25.8 ± 8.6 × 10^7^) than in the 24–96-h G-CSF-treated group (17.7 ± 7.9 × 10^7^, *p* < 0.0001) and the non-G-CSF group (14.4 ± 8.0 × 10^7^, *p* < 0.0001). The ratio of DCs to apheresed monocytes was significantly higher in the 24-h (20.0% ± 7.5%) than in the 24–96-h G-CSF-treated group (14.1% ± 7.5%; *p* = 0.002) and the non-G-CSF-treated group (15.2 ± 6.9%; *p* = 0.004) (Figure 3b).

The intensity of the yielded DCs with CD11c was increased in the 24-h G-CSF-treated group compared with the non-treated group (*p* = 0.004), while CD14 was expressed to a lower degree in the 24-h G-CSF-treated group (*p* = 0.040). CD80^+^ DCs in the 24-h G-CSF-treated group were compared with the 24–96-h G-CSF-treated and non-treated groups (*p* = 0.015 and *p* = 0.012, respectively) (Figure 3c).

The correlation between peripheral blood monocyte counts before (410 ± 195/μL) and 24 h after (586 ± 238/μL) the subcutaneous administration of rhG-CSF (75 μg filgrastim) in 40 available cases was statistically significant (Figure 4a). CD34^+^ cells were present to a lower degree in 20 available cases treated with a low-dose exposure of G-CSF for 24 h (*p* = 0.0001), being 0.401 ± 0.273/μL and 0.169 ± 0.056/μL (*p* < 0.001), respectively, following the administration of rhG-CSF (75 μg filgrastim) (Figure 4b).

### 3.4. The Acquisition of WT1-specific CTLs

WT1-specific CTLs were detected using WT1-tetramer analysis (Table 1). The distribution of G-CSF-primed and non-primed patients per cancer type was not statistically significant (*p* = 0.868). The administered DCs were not significantly different between G-CSF-primed (9.13 ± 2.42 × 10^7^) and non-primed (8.64 ± 2.81 × 107) groups (*p* = 0.320). The induction ratio of WT1-CTLs (tetramer^+^/CD8^+^T cells) by tetramer analysis was 46/54 patients (85.2%), while a significant increase was found in 36/39 (92.3%) patients treated with DC vaccine in vivo primed with G-CSF compared with that of 10/15 (66.7%) patients without priming (*p* = 0.019). The median percentage of WT1-CTLs after one course of DC vaccination was 0.05% (range: 0.01%–0.33%) in the G-CSF-treated group and 0.06% (range: 0.01%–0.38%) in the non-G-CSF group (*p* = 0.712). The median growth rate of WT1-CTLs between the 1^st^ and 7^th^ sessions was 0.03% (range: −0.02%–0.31%) and 0.02% (range: −0.03%–0.38%) in the G-CSF-treated and non-G-CSF groups, respectively (*p* = 0.462). These findings suggest the immune effectiveness of in vivo G-CSF-primed DC vaccines by an increase in the ratio of acquired immunity as detected by the positive levels of WT1-specific CTLs.

## 4. Discussion

In the era of the field of personalized cancer therapies, feasible, well-tolerated, and promising immunologic and multimodal therapies for various types of cancers are expected. Quality-verified DC vaccines are required to achieve the expected level of effectiveness [32]. The reliable and reproducible manufacturing of DC vaccines is necessary for multicenter clinical applications and trials.

We retrospectively evaluated the clinical data from 108 apheresis sessions performed in 90 patients with advanced cancers. The conventional procedure for obtaining monocyte-derived DCs was validated by the phenotype and viability of the yielded cells (Figure 1 and Figure 2). Administration of 75–100 μg of rhG-CSF 24–96 h prior to apheresis for neutropenia caused by chemotherapy was used to stimulate the proliferation of myeloid precursors and accelerate the release of neutrophils. Besides the effect of rhG-CSF in relation to the number of collected monocytes, our study revealed a significant increase in the DC yield and DC/monocyte-ratio in the rhG-CSF-treated groups within 24 h, suggesting favorable hematological conditions for both the quantitative and qualitative yield of DCs. The number of monocytes increased by >50% compared with that 16 h–18 h prior to administration with 75 μg filgrastim, but CD34^+^ stem cells were not mobilized by the dose of rhG-CSF and rather decreased in numbers (Figure 4), implying that mobilized CD34^+^ monocytes did not affect the yield of DCs. rhG-CSF increases the peripheral blood monocyte count due to mediation via G-CSF receptors that are present on at least some subsets of monocytes [33,34]. Overexposure of mobilizing doses of G-CSF may induce T cell tolerance via direct and indirect mechanisms involved in regulatory T cells, regulatory DCs, myeloid-derived suppressor cells, and CD34^+^ monocytes. G-CSF also selectively induces a subset of plasmacytoid DCs that leads to the polarization of T cells from the Th1 to Th2 phenotype [35]. High-dose treatment with G-CSF (10 μg/kg daily for 5 days) in vivo resulted in the increased number of mobilized CD34^+^ cells and DCs [36]. While the administration of 75 μg filgrastim 16–18 h prior to apheresis did not mobilize CD34^+^ cells, rather indicating an optimized condition for ex vivo induction of DCs from monocytes that led to an increase in the ratio of patients who could acquire immunity against cancer-associated antigens (Table 1). These findings suggested that low-dose G-CSF and short periods of G-CSF exposure responded via different mechanisms to maintain homeostasis from that of high-dose administration, leading to a tolerable immunological environment.

mDCs express cell surface molecules necessary for antigen presentation. CD83, CD86, and HLA-DR, which is higher than that on immature DCs, were found to stimulate T cells in vitro in almost all mDCs [37]. The expression of CD197 (C-C chemokine receptor type 7) was also shown to increase with maturation and induced chemotaxis associated with macrophage inflammatory protein 3β [38]. mDCs manufactured using GM-CSF and IL-4 primed with OK-432 for clinical use expressed the HLA^−^ABC^+^DR^+^CD40^+^CD80^+^CD86^+^CD197^+^ phenotype, harboring bioactive functions that could be useful in providing personalized vaccines for cancer immunotherapy [32]. mDCs primed with G-CSF within 24 h consisted more prominently of a CD11c^+^CD14^−^ population than untreated DCs. The HLA-related molecule CD80 was detected at higher levels in mDCs primed with G-CSF within 24 h (Figure 3c). The CD80 antigen is the ligand of both CD28 and CTLA-4 and acts as one of the key molecules that stimulate T cells [39,40,41,42,43,44]. Despite no quantitative increase in the numbers of CTLs, the clinical evaluation revealed a higher ratio of the induced WT1-CTLs using low-dose rhG-CSF priming in vivo (Table 1). The higher level of expression of CD80 on DCs subjected to rhG-CSF priming is suggested to provide an antigen-presenting ability of their DCs to T cells, linking the acquisition of immunity with amplification of the CD80 antigen on DCs. As much as CD11c^+^CD14^−^CD80^+^ DCs primed with G-CSF would be expected to promote GCTP-validated manufacture of DC vaccines for clinical use, further investigation is necessary to clarify their underlying molecular mechanisms, including the effects of adhesion molecules in DCs from monocytes primed with G-CSF.

## 5. Conclusions

A low dose of rhG-CSF exposure for 16–18 h in vivo was useful to increase the yield of CD11c^+^CD14^−^CD80^+^ DCs. The DC vaccines primed with a low dose of rhG-CSF in vivo induced a higher DC/monocyte ratio in patients with antigen-specific CTLs after DC vaccination than that which would be expected for the development of immunogenicity for cancer immunotherapy. This is the first report to reveal the clinical effectiveness of acquired immunity using G-CSF-primed DC vaccines. Notably, future research should verify whether G-CSF-primed DC vaccines would affect clinical outcomes.

## 6. Patents

S.S., T.K. (Terutsugu Koya), and Y.H. are inventors of the patent for the manufacturing of a DC vaccine using G-CSF (PCT/JP/2014/053676). H.S. is the inventor of the WT1 patent (PCT/JP02/02794 and PCT/JP04/16336).

## Figures and Tables

**Figure 1 vaccines-07-00120-f001:**
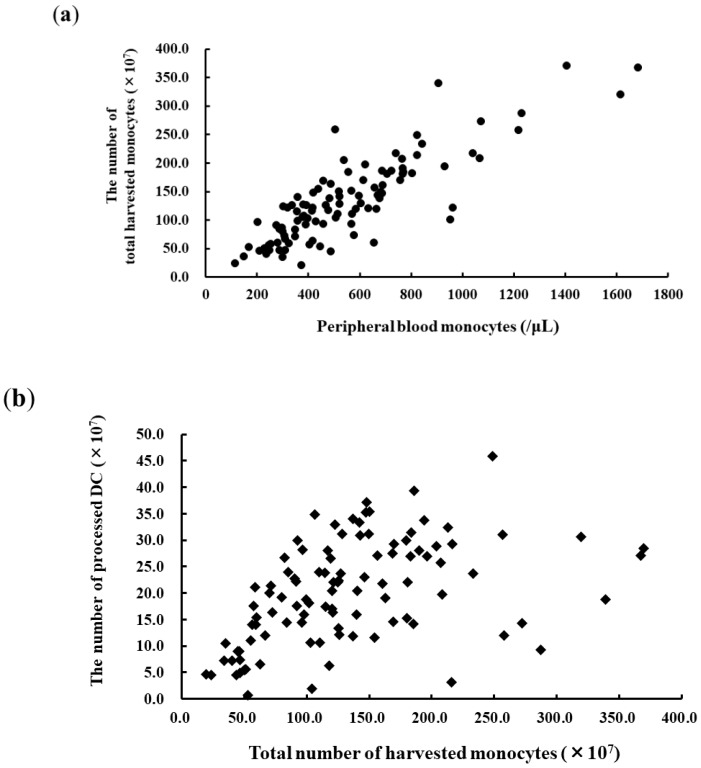
The yield of dendritic cells (DCs) depending on the number of apheresed monocytes. (**a**) Correlation between peripheral blood monocyte counts and the number of monocytes harvested by apheresis (*r* = 0.828, *p* < 0.0001, *n* = 108). (**b**) The number of DCs that could be harvested was dependent on the number of apheresed monocytes (*r* = 0.435, *p* < 0.0001, *n* = 108). Mean number of DCs, 20.2 × 10^7^; DC/apheresed monocyte-ratio, 17.0% ± 7.7%.

**Figure 2 vaccines-07-00120-f002:**
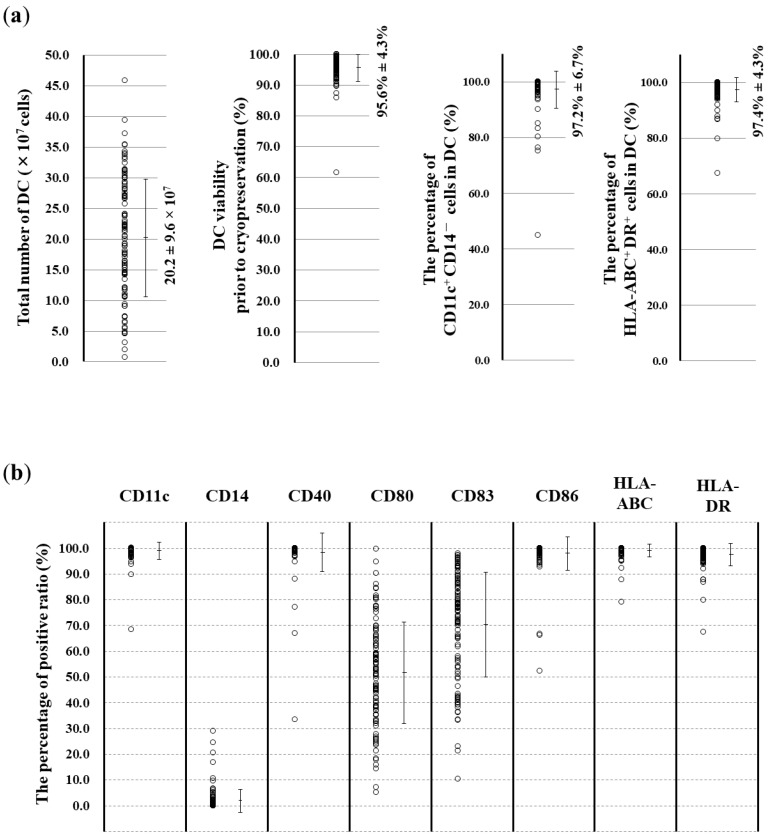
Analysis of the dendritic cell (DC) yield (*n* = 108). (**a**) The number of DCs and the cell viability prior to cryopreservation was 20.2 ± 9.6 × 10^7^ and 95.6% ± 4.3%, respectively. The percentages of the yielded DCs with the CD11c^+^CD14^−^ and human leucocyte antigen (HLA)-ABC^+^DR^+^ phenotypes, representative of DCs, were 97.2% ± 6.7% and 97.4% ± 4.3%, respectively. Error bars on the dot plots represent the mean ± SD. (**b**) Single-color flow cytometric analysis of the mDCs (% of positive ratio): CD11c (99.0 ± 3.3); CD14 (1.9 ± 4.6); CD40 (98.3 ± 7.5); CD80 (51.6 ± 19.6); CD83 (70.3 ± 20.4); CD86 (98.0 ± 6.4); HLA-ABC (99.1 ± 2.6); and HLA-DR (97.8 ± 4.3). Mean positive ratios in the panels for CD11c, CD40, CD86, HLA-ABC, and HLA-DR were >90%. Error bars on the dot plots represent the mean ± SD.

**Figure 3 vaccines-07-00120-f003:**
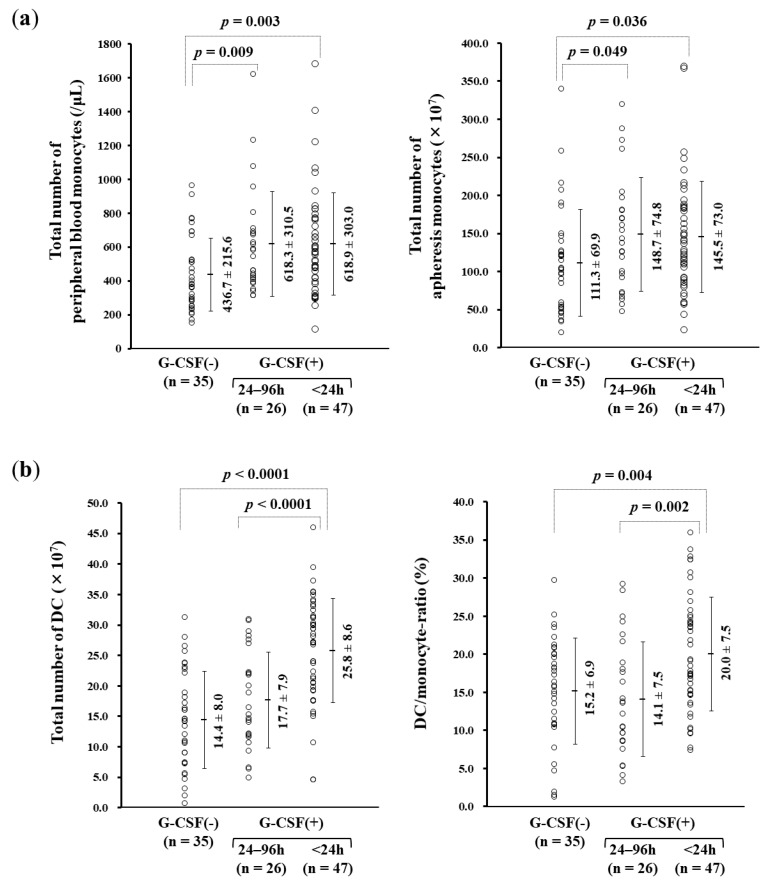
Effect of recombinant human granulocyte colony-stimulating factor (rhG-CSF) on the number of monocytes and the monocyte-derived dendritic cell (DC) yield. (**a**) The numbers of both peripheral and apheresed monocytes were higher in G-CSF-treated groups than in the non-G-CSF group. The number of peripheral blood monocytes (/μL) were as follows: G-CSF(−), 436.7 ± 215.6; 24–96 h, 618.3 ± 310.5; and <24 h, 618.9 ± 303.0 (mean ± SD). The total numbers of apheresis monocytes (×10^7^) were as follows: G-CSF(−), 111.3 ± 69.9; 24–96 h, 148.7 ± 74.8; and <24 h, 145.5 ± 73.0 (mean ± SD). (**b**) The DC yield and the ratio of DCs to apheresed monocytes (DC/monocyte-ratio) were significantly higher in the 24-h than in the 24–96-h G-CSF-treated and the non-G-CSF-treated groups. The total number of DC (×10^7^) were as follows: G-CSF(−), 14.4 ± 8.0; 24–96 h, 17.7 ± 7.9; and <24 h, 25.8 ± 8.6 (mean ± SD). DC/monocyte-ratio (%) were G-CSF(−), 15.2 ± 6.9; 24–96 h, 14.1 ± 7.5; and <24 h, 20.0 ± 7.5 (mean ± SD). (**c**) Effect of rhG-CSF on the monocyte-derived DC phenotype. The intensity of the yielded DCs with CD11c and CD80 was increased, while CD14 expression occurred to a lower degree in the 24-h compared with the 24–96-h G-CSF-treated and the non-treated groups. Positive ratio (%) of CD11c were 98.9 ± 1.9, 97.7 ± 6.2, and 99.9 ± 0.8 in groups G-CSF(−), 24–96 h, and <24 h (mean ± SD), respectively. The ratio of CD14^+^ was 2.0 ± 3.4, 3.7 ± 8.1, and 0.8 ± 1.3, respectively, and that of CD80^+^ was 47.1 ± 20.2, 46.5 ± 20.5, and 57.6 ± 17.0, respectively.

**Figure 4 vaccines-07-00120-f004:**
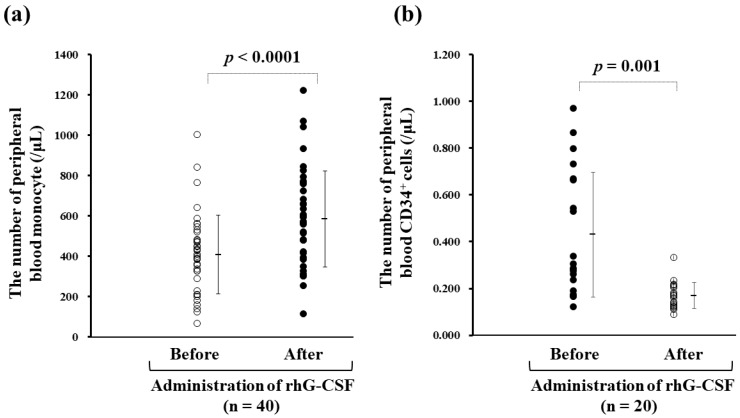
Correlation between peripheral blood monocyte counts before and within 24 h after administration of recombinant human granulocyte colony-stimulating factor (rhG-CSF) (75 μg filgrastim). (**a**) The peripheral blood monocyte counts before and 16–18 h after administration of rhG-CSF were 410 ± 195/μL and 586 ± 238/μL (n = 40), respectively, indicating a 1.5-fold increase. (**b**) CD34^+^ stem cells did not increase 24 h after rhG-CSF administration (before, 0.401 ± 0.273/µL; after 0.169 ± 0.056/µL (*n* = 20).

**Table 1 vaccines-07-00120-t001:** The acquisition of Wilms’ tumor 1 (WT1) antigen-specific cytotoxic T cells in patients with human leucocyte antigen class-A*24:02.

G-CSF (75 μg of Filgrastim) 16–18 h Prior to Apheresis (*N* = 54)	Treated	Untreated	*p*-Value ^¶^
	39	15	
Number of DCs in 1 course (mean ± SD)	9.13 ± 2.42 × 10^7^	8.64 ± 2.80 × 10^7^	*p* = 0.320
Induction ratio of WT1-CTLs * after one course (%) ^#^	36 (92.3%)	10 (66.7%)	*p* = 0.019
Median level of WT1-CTLs * after one course (range)	0.05% (0.01–0.33)	0.06% (0.01–0.38)	*p* = 0.712
Median growth rate of WT1-CTLs * between the 1st and 7th session (range)	0.03% (−0.02–0.31)	0.02% (−0.03–0.38)	*p* = 0.462

* WT1 antigen-specific cytotoxic T cells; ^#^ tetramer^+^/CD8^+^T-cells > 0.2%; ^¶^ Mann–Whitney *U*-test. G-CSF, granulocyte colony-stimulating factor; CTL, cytotoxic T cells.

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
