# Peer review of "In Vivo Administration of Recombinant Human Granulocyte Colony-Stimulating Factor Increases the Immune Effectiveness of Dendritic Cell-Based Cancer Vaccination"

_vaccines, 2019, doi:10.3390/vaccines7030120_

Round 1

Reviewer 1 Report

Title could be more precise. Consider changing „granulocyte colony-stimulating factor”
to: „recombinant human granulocyte colony-stimulating factor”
or to „filgrastim” (suggestion). Line "39". Consider changing "chemotherapy" to  more general term: „systemic therapy” as targeting immune checkpoints is not „classical chemotherapy”. Lines "42 - 44". Can such APC activation be sufficient / effective when cancer cells inhibit lymphocyte activity through the PD-1 receptor? Please expand this topic. Line 75 - "(...) December 2, 2008; 2704, April 8, 2014)" - why were 2 dates given? Lines 78-80. The sum of numbers for each site (point of origin) is 108, not 90. Please explain. Line 89. Does it mean that 39 out of 43 received filgrastim? Please explain. Moreover, there is no justification for adopted method of patients allocation to both groups. Line 103. Are these conditions equivalent (when it comes to receiving the final number of cells and their viability)? Line 140.Was it supposed to be 100,000? Lines 143-152. Please explain why both parametric and non-parametric tests were used. Line 188. Please indicate how it was administered (intravenously, subcutaneously etc.).  Please also explain what leukocyte count was taken as the cut-off for GCSF administration (e.g. CTCAE grade). Was the leukopenia the only indication? How about the chemotherapy regimens and their risk of febrile neutropenia (administration of GCSF as primary prophylaxis). Table 1. Please consider moving „G-CSF (75 μg of filgrastim) 16–18 h prior to apheresis (N = 54)” above the „+” and „-" section or change „+” and „-" to more descriptive „treated” / „untreated”. Conclusions. It's worth mentioning that future research should verify whether G-CSF-primed DC vaccines would affect clinical outcomes.

Author Response

Reviewer 1

Comments and Suggestions for Authors

Title could be more precise. Consider changing „granulocyte colony-stimulating factor” 
to: „recombinant human granulocyte colony-stimulating factor” 
or to „filgrastim” (suggestion).

Answer) We really appreciate your kind comments giving us your excellent suggestion.

We changed the title as reviewer’s suggestion as “In vivo administration of recombinant human granulocyte colony-stimulating factor increases the immune effectiveness of dendritic cell-based cancer vaccination”.

Line "39". Consider changing "chemotherapy" to more general term: „systemic therapy” as targeting immune checkpoints is not „classical chemotherapy”.

Answer) We changed “systemic therapy” as reviewer’s comment.

Lines "42 - 44". Can such APC activation be sufficient / effective when cancer cells inhibit lymphocyte activity through the PD-1 receptor? ã€€Please expand this topic.

Answer) We add the sentences and literature, lines 45–50.

DCs expressing both PD-L1 and PD-1 can virtually interact with any PD-1 and PD-L1-positive cells, respectively, with suppressive activity on CD4+ and CD8+ T cells and promotive expression of CD4+CD25+FoxP3+ regulatory T cells [8]. Interference with the PD-1/PD-L pathway can increase the immunostimulatory properties of the DCs to activate T cells [8]. It is possible that the immune checkpoints on the DCs would interfere with the response of antitumor immunity.  

Versteven, M; Van den Bergh, JMJ; Marcq, E; Smits, ELJ; Van Tendeloo, VFI; Hobo, W; Lion, E. Dendritic cells and programmed death-1 blockade: A joint venture to combat cancer. Front. Immunol. 2018, 9, 394.

Line 75 - "(...) December 2, 2008; 2704, April 8, 2014" - why were 2 dates given?

Answer) The reason was added in the text, lines 88–89, as "December 2, 2008 (initial clinical study) and 2704, April 8, 2014 (consecutively followed approval of advanced medical care in Japan)".

Lines 78-80. The sum of numbers for each site (point of origin) is 108, not 90. Please explain.

Answer) We added the exact statement as “A total of 108 apheresis sessions were conducted for 90 patients with advanced cancers, such as breast (17), colon/rectum (19), lung (12), ovary (11), pancreatic (33), and stomach (16) cancers, including repeated sessions in 18 patients;”, in lines 92–94.

Line 89. Does it mean that 39 out of 43 received filgrastim? Please explain. Moreover, there is no justification for adopted method of patients allocation to both groups.

Answer) We added the words to understand exactly as lines 102–105.

“The first 15 patients were consecutively enrolled into the group without rhG-CSF priming, and then the next 43 patients were enrolled in the group that was treated with rhG-CSF 16–96 h before the apheresis session. Thirty-nine out of 43 patients who were primed with rhG-CSF (75 μg filgrastim) 16–18 h prior to apheresis. “

Line 103. Are these conditions equivalent (when it comes to receiving the final number of cells and their viability)?

Answer) We added the sentence in lines 111–112. The final number of viable DCs was equivalently aliquoted as 1 × 107 in each lot.

Line 140.Was it supposed to be 100,000?

Answer) We changed correctly as 100,000 in line 156.

Lines 143-152. Please explain why both parametric and non-parametric tests were used.

Answer) We added the meaning of the analyses in the paragraph 2.5 Statistical Analysis with red colored words, lines 160–170.

Line 188. Please indicate how it was administered (intravenously, subcutaneously etc.).  

Please also explain what leukocyte count was taken as the cut-off for GCSF administration (e.g. CTCAE grade). Was the leukopenia the only indication? How about the chemotherapy regimens and their risk of febrile neutropenia (administration of GCSF as primary prophylaxis).

Answer) Lines 211–214: The cut-off for rhG-CSF administration was defined either as neutrophil count lower than 500/μL or counts lower than 1,000/μL with a risk of fever. There were no cases with adverse effects due to the subcutaneous administration of rhG-CSF during the chemotherapy regimens.  

Table 1. Please consider moving „G-CSF (75 μg of filgrastim) 16–18 h prior to apheresis (N = 54)” above the „+” and „-" section or change „+” and „-" to more descriptive „treated” / „untreated”.

Answer) We changed the character in Table 1 as reviewer’s advice.

Conclusions. It's worth mentioning that future research should verify whether G-CSF-primed DC vaccines would affect clinical outcomes. 

Answer) We added the reviewer’s recommended statement in 5. Conclusions.

Reviewer 2 Report

Shimodaira et al show that DC vaccine for immunotherapy primed with low-dose of recombinant human granulocyte colony stimulating factor results in higher immunogenicity. Study has clinical significance, and manuscript is well-written. Below are my comments.

Please give a brief overview of early and ongoing DC vaccine trials and cancer models. It would be helpful for the readers if authors can discuss other recent studies with DC for cancer. Response rates to DC vaccination vary among cancer types with most studies showing response rates between 10 and 15%. Please explain the benefit associated with dendritic cell-based cancer vaccination for metastasis. Please discuss the rationale for DC vaccination in the adjuvant treatment of cancer? The clinical benefit of monotherapy DC vaccination for patients with metastatic disease is probably limited. However, the ultimate role for vaccines may lie in the combination with other modalities. Please discuss the combination of DC vaccination and other modalities for the treatment of cancer. We know from published study that granulocyte colony-stimulating factor mobilizes more dendritic cell subsets In vivo, Stem Cells. 2006 Jul;24(7):1789-97. Authors should explain how current study is novel and adding to the existing knowledge of scientific community.

Author Response

Reviewer 2

Please give a brief overview of early and ongoing DC vaccine trials and cancer models. It would be helpful for the readers if authors can discuss other recent studies with DC for cancer.

Answer) We really appreciate your kind comments giving us your excellent suggestion.

We added our previous studies about DC vaccines targeting WT1 in lines for readers.

Lines 59–62: DC vaccines targeting Wilms’ tumor 1 (WT1) have been previously shown to be safe and feasible with few adverse reactions in patients with advanced cancer, including colorectal cancer [12], pancreatic cancer [13,14], lung cancer [15], high-grade glioma [16], and pediatric cancer [17,18].  

Shimodaira, S.; Sano, K.; Hirabayashi, K.; Koya, T.; Higuchi, Y.; Mizuno, Y.; Yamaoka, N.; Yuzawa, M.; Kobayashi, T.; Ito, K.; et al. Dendritic cell-based adjuvant vaccination targeting Wilms’ tumor 1 in patients with advanced colorectal cancer. Vaccines (Basel) 2015, 3, 1004–1018. Kobayashi, M.; Shimodaira, S.; Nagai, K.; Ogasawara, M.; Takahashi, H.; Abe, H.; Tanii, M.; Okamoto, M.; Tsujitani, S.; Yusa, S.; et al. Prognostic factors related to add-on dendritic cell vaccines on patients with inoperable pancreatic cancer receiving chemotherapy: a multicenter analysis. Cancer Immunol. Immunother. 2014, 63, 797–806. Koido S.; Homma S.; Okamoto M.; Takakura K.; Mori M.; Yoshizaki S.; Tsukinaga S.; Odahara S.; Koyama S.; Imazu H.; et al. Treatment with chemotherapy and dendritic cells pulsed with multiple Wilms' tumor 1 (WT1)-specific MHC class I/II-restricted epitopes for pancreatic cancer. Clin. Cancer Res. 2014, 20, 4228–4239. Takahashi, H.; Shimodaira, S.; Ogasawara, M.; Ota, S.; Kobayashi, M.; Abe, H.; Morita, Y.; Nagai, K.; Tsujitani, S.; Okamoto, M.; et al. Lung adenocarcinoma may be a more susceptive subtype to a dendritic cell-based cancer vaccine than other subtypes of non-small cell lung cancers: a multicenter retrospective analysis. Cancer Immunol. Immunother. 2016, 65, 1099–1111. Sakai K.; Shimodaira S.; Maejima S.; Udagawa N.; Sano K.; Higuchi Y.; Koya T.; Ochiai T.; Koide M.; Uehara S.; et al. Dendritic cell-based immunotherapy targeting Wilms' Tumor 1 (WT1) in patients with relapsed malignant glioma. J. Neurosurg. 2015, 123, 989–997. Saito, S.; Yanagisawa, R.; Yoshikawa, K.; Higuchi, Y.; Koya, T.; Yoshizawa, K.; Tanaka, M.; Sakashita, K.; Kobayashi, T.; Kurata, T.; et al. Safety and tolerability of allogeneic dendritic cell vaccination with induction of WT1-specific T cells in a pediatric donor and pediatric patient with relapsed leukemia: A case report and review of the literature. Cytotherapy. 2015, 17, 330–335. Shimodaira, S.; Hirabayashi, K.; Yanagisawa, R.; Higuchi, Y.; Sano, K.; Koizumi, T. Dendritic cell-based cancer immunotherapy targeting Wilms’ Tumor 1 for pediatric cancer. In: Wilms Tumor; van den Heuvel-Eibrink, M.M., Ed; Codon Publications: Brisbane, Australia, 2016; Chapter 8.

Response rates to DC vaccination vary among cancer types with most studies showing response rates between 10 and 15%. Please explain the benefit associated with dendritic cell-based cancer vaccination for metastasis.

Answer) We explained the efficacy of DC vaccination as showing the references in lines 62–65:

The efficacy of DC-based immunotherapy is more evidently demonstrated by the delayed separation of the survival curve with a benefit in terms of prolonged overall survival rather than conventional evaluation approaches such as the use of the response rates [13,15,19].

Hoos, A. Evolution of end points for cancer immunotherapy trials. Ann. Oncol. 2012, 8, 47–52.

Please discuss the rationale for DC vaccination in the adjuvant treatment of cancer? The clinical benefit of monotherapy DC vaccination for patients with metastatic disease is probably limited. However, the ultimate role for vaccines may lie in the combination with other modalities. Please discuss the combination of DC vaccination and other modalities for the treatment of cancer.

Answer) We added the following sentence as “We administered add-on DC vaccination in combination with conventional chemotherapy for each patient.” in lines 86–87.

We performed neither adjuvant setting nor monotherapy.

We know from published study that granulocyte colony-stimulating factor mobilizes more dendritic cell subsets In vivoStem Cells. 2006 Jul;24(7):1789-97. Authors should explain how current study is novel and adding to the existing knowledge of scientific community. 

Answer) Lines 298–300: We sited the article and added the following sentence.

“High-dose treatment with G-CSF (10 μg/kg daily for 5 days) in vivo resulted in the increased number of mobilized CD34+ cells and DCs [36].”, in lines following the sentences with novel and adding knowledge of scientific community.

Shaughnessy, P.J.; Bachier, C.; Lemaistre, C.F.; Akay, C.; Pollock, B.H.; Gazitt, Y. Granulocyte colony-stimulating factor mobilizes more dendritic cell subsets than granulocyte-macrophage colony-stimulating factor with no polarization of dendritic cell subsets in normal donors. Stem Cells. 2006, 24, 1789–1797.

Reviewer 3 Report

Dear authors, I think that the paper is well-written and of interest. I would suggest an improvement of figures 2-3-4. I really had difficulties in reading and understanding them. Would you consider to make them bigger and simplify them so that they may appear more readable?

Thanks

Author Response

Reviewer 3

Dear authors, I think that the paper is well-written and of interest. I would suggest an improvement of figures 2-3-4. I really had difficulties in reading and understanding them. Would you consider to make them bigger and simplify them so that they may appear more readable?

Answer) We really appreciate your kind comments giving us your excellent suggestion.

We provide the Figures revised clearly and easily understandable for readers following the reviewer's kind advise. The complex data in the Figures were also moved to legends of the Figures.

We also attached high-resolution Figures with 300dpi in the ZIP holder.